# SIAMESE SURVIVAL ANALYSIS WITH COMPETING RISKS

## ABSTRACT

Survival Analysis (time-to-event analysis) in the presence of multiple possible adverse events, i.e., competing risks, is a challenging, yet very important problem in medicine, finance, manufacturing, etc. Extending classical survival analysis to competing risks is not trivial since only one event (e.g. one cause of death) is observed and hence, the incidence of an event of interest is often obscured by other related competing events. This leads to the nonidentifiability of the event times distribution parameters, which makes the problem significantly more challenging. In this work we introduce Siamese Survival Prognosis Network, a novel Siamese Deep Neural Network architecture that is able to effectively learn from data in the presence of multiple adverse events. The Siamese Survival Network is especially crafted to issue pairwise concordant time-dependent risks, in which longer event times are assigned lower risks. Furthermore, our architecture is able to directly optimize an approximation to the C-discrimination index, rather than relying on well-known metrics of cross-entropy etc., and which are not able to capture the unique requirements of survival analysis with competing risks. Our results show consistent performance improvements on a number of publicly available medical datasets over both statistical and deep learning state-of-the-art methods.

## 1 INTRODUCTION

### 1.1 MOTIVATION

Survival analysis is a method for analyzing data where the target variable is the time to the occurrence of a certain adverse event. Competing risks is an extension to the classical survival analysis in which we distinguish between multiple possible adverse events. The application of survival analysis are numerous and include medicine, finance, manufacturing etc. While our work is applicable to all these domains, we will mainly focus on its application to medicine, where competing risk analysis has emerged in recent years as an important analysis and predictive tool in medicine (Glynn & Rosner (2005); Wolbers et al. (2009); Satagopan et al. (2004)), where an increasingly aging population is suffering from multiple comorbidities. For instance, studies in cardiology often record the time to multiple disease events such as heart attacks, strokes, or hospitalization. Competing risks methods allow for the analysis of the time to the first observed event and the type of the first event. They are also relevant if the time to a specific event is of primary interest but competing events may preclude its occurrence or greatly alter the chances to observe it.

### 1.2 RELATED WORKS

Previous work on classical survival analysis has demonstrated the advantages of deep learning over statistical methods (Luck et al. (2017); Katzman et al. (2016); Yousefi et al. (2017)). Cox proportional hazards model (Cox (1972)) is the basic statistical model for Survival Analysis. One limitation of Cox PH model is that its time dependent risk function is the product of a linear covariate function and a time dependent function. Katzman et al. (2016) replaced the linear covariate function with a feed-forward neural network and demonstrated performance improvements. They used a neural network to learn a representation which is then used as the input for the Cox PH model. However, the problem of competing risks is much less studied, and the literature is based on classical methods based on statistics (the Fine Gray model (Fine & Gray (1999))), classical machine learning (Ran-

dom Survival Forest (Ishwaran et al. (2008; 2014))), multi-task learning (Alaa & van der Schaar (2017)), etc. A challenge of the existing competing risks models is that they do not scale to datasets where many patients and many covariates. To address this challenge, we are proposing to use a deep learning architecture. However, designing a deep learning architecture that is able to handle survival analysis with competing risks is challenging because it needs to optimize the time-dependent discrimination index, which is not straightforward (see next section).

## 1.3 CONTRIBUTIONS

In both machine learning and statistics it is common to develop predictive models and compare them in terms of the area under the receiver operating characteristic curve or the time-dependent discrimination index (in the survival analysis literature). The equivalence of the two metrics was established in Heagerty & Zheng (2005). Numerous works on supervised learning (Cortes & Mohri (2004); Yan et al. (2003); Luaces et al. (2007); Chen et al. (2013); Agarwal et al. (2005); Mayr & Schmid (2014); Mayr et al. (2016); Schmid et al. (2016)) have shown that training the models directly optimizing the AUC can lead to much better out-of-sample (generalization) performance (in terms of AUC) rather than optimizing the error rate (or the accuracy). In this work, we adopt this idea to survival analysis with competing risks. We develop a novel Siamese feed-forward neural network (Bromley et al. (1994)) which is designed to optimize concordance and specifically, the time-dependent discrimination index (Antolini et al. (2005)) (which is able to take competing risks into account). This is achieved by estimating risks in a relative fashion, meaning, the risk for the true event of a patient (i.e. the event which actually took place) must be higher than: all other risks for the same patient and the risks for the same true event of other patients that experienced it at a later time. Furthermore, the risks for all the causes are estimated jointly in an effort to generate a shared representation that captures the latent structure of the data and to estimate cause-specific risks. Because our neural network issues a joint risk for all competing events, our architecture needs to compare different risks for the different events at different times and arrange them in a concordant fashion (earlier time means higher risk for any pair of patients).

Unlike previous Siamese neural networks architectures (Chopra et al. (2005); Bromley et al. (1994); Wang et al. (2017)) which were developed for different purposes such as learning the pairwise similarity between different inputs, our architecture aims to maximize the gap between output risks among the different inputs. Instead of learning a representation that captures the similarities between the inputs, we learn a representation that generates the highest possible difference between the outputs. We overcome the discontinuity problem of the above metric by introducing a continuous approximation of the time-dependent discrimination function. This approximation is only evaluated at the survival times observed in the dataset. However, training a neural network only over the observed survival times can lead to a model that does not generalize well for other times, which can lead to poor out of sample performance (in terms of discrimination index computed at different times). To overcome this problem, we add a loss term (to the loss function) which for any pair of patients, forces the survival curve of the right patient (longer survival time) to be lower than the survival curve of the left patient (shorter survival time) up to the event time of the left patient and as a result, improves the generalization capabilities of our algorithm.

In addition, we address the competing risks nonidentifiability problem (which arises from the inability to estimate the true cause-specific survival curves from the empirical data (Tsiatis (1975))) by generating concordant risks as opposed to true cause-specific survival curves. By avoiding the estimation of the true cause-specific survival curves, we are able to avoid the nonidentifiability problem.

We report modest yet statistically significant improvements over the state-of-the-art methods on survival analysis with competing risks on both synthetic as well as on real medical data. However, we wish to remind the reader that our focus is on healthcare were even minor gains are important because of the potential to save lives. For instance, there are 72809 patients in the SEER dataset we used. A performance improvement even as low as 0.1% has the potential to save lives and therefore should not be disregarded.

## 2 PROBLEM FORMULATION

We consider a dataset $\mathcal{H}$ comprising of time-to-event information about $N$ subjects who are followed up for a finite amount of time. Each subject (patient) experiences an event $D \in \{0, 1, .., M\}$, where $D$ is the event type. $D = 0$ means the subject is censored (lost in follow-up or study ended). If $D \in \{1, .., M\}$, then the subject experiences one of the events of interest (for instance, subject develops cardiac disease). We assume that a subject can only experience one of the above events and that the censorship times are independent of them (Lim et al. (2010); Lambert et al. (2010); Satagopan et al. (2004); Fine & Gray (1999); Crowder (2001); Gooley et al. (1999); Tsiatis (1975)). $T$ is defined as the time-to-event, where we assume that time is discrete $T \in \{t_1, ..., t_K\}$ and $t_1 = 0$ ($t_i = 0$ denotes the elapsed time since $t_1$). Let $\mathcal{H} = \{T_i, D_i, x_i\}_{i=1}^N$, where $T_i$ is the time-to-event for subject $i$, $D_i$ is the event experienced by the subject $i$ and $x_i \in \mathbb{R}^S$ are the covariates of the subject (the covariates are measured at baseline, which may include age, gender, genetic information etc.).

The Cumulative Incidence Function (CIF) (Fine & Gray (1999)) computed at time $t$ for a certain event $D$ is the probability of occurrence of a particular event $D$ before time $t$ conditioned on the covariates of a subject $x$, and is given as $F(t, D|x) = Pr(T \leq t, D|x)$. The cumulative incidence function evaluated at a certain point can be understood as the risk of experiencing a certain event before a specified time.

In this work, our goal is to develop a neural network that can learn the complex interactions in the data and is particularly suited to this setting of competing risks survival analysis. We need to decide the loss function to use and the architecture of the neural network. Time-dependent discrimination index is the most commonly used metric for evaluating models in survival analysis (Antolini et al. (2005)). There are many works in the supervised learning literature that have shown that approximating the area under the curve (AUC) directly and training a classifier leads to better generalization performance in terms of the AUC (see e.g. Cortes & Mohri (2004); Yan et al. (2003); Luaces et al. (2007); Chen et al. (2013); Agarwal et al. (2005); Mayr & Schmid (2014); Mayr et al. (2016); Schmid et al. (2016)). However, these ideas were not explored in the context of survival analysis with competing risks. We will follow the same principles to construct an approximation of the time-dependent discrimination index to train our neural network. We first describe the time-dependent discrimination index below.

Consider an ordered pair of two subjects $(i, j)$ in the dataset. If the subject $i$ experiences event $m$, i.e., $D_i \neq 0$ and if subject $j$'s time-to-event exceeds the time-to-event of subject $i$, i.e., $T_j > T_i$, then the pair $(i, j)$ is a comparable pair. The set of all such comparable pairs is defined as the comparable set for event $m$, and is denoted as $X^m$.

A model outputs the risk of the subject $x$ for experiencing the event $m$ before time $t$, which is given as $R^m(t, x) = F(t, D = m|x)$. The time-dependent discrimination index for a certain cause $m$ is the probability that a model accurately orders the risks of the comparable pairs of subjects in the comparable set for event $m$. The time-dependent discrimination index (Antolini et al. (2005)) for cause $m$ is defined as

$$C^{td}(m) = \frac{\sum_{k=1}^{K} AUC^m(t_k) w^m(t_k)}{\sum_{k=1}^{K} w^m(t_k)} \tag{1}$$

where

$$AUC^m(t_k) = Pr\{R^m(t_k, x_i) > R^m(t_k, x_j)|T_i = t_k, T_j > t_k, D_i = m\} \tag{2}$$

$$w^m(t_k) = Pr\{T_i = t_k, T_j > t_k, D_i = m\} \tag{3}$$

The discrimination index in (1) cannot be computed exactly since the distribution that generates the data is unknown. However, the discrimination index can be estimated using a standard estimator described next (the estimator takes as input the risk values associated with subjects in the dataset). Antolini et al. (2005) defines the estimator for (1) as

$$\hat{C}^{td}(m) = \frac{\sum_{j=1}^{N} \sum_{i=1}^{N} \mathbf{1}\{R^m(T_i, x_i) > R^m(T_i, x_j) \cdot \mathbf{1}\{T_j > T_i, D_i = m\}\}}{\sum_{j=1}^{N} \sum_{i=1}^{N} \mathbf{1}\{T_j > T_i, D_i = m\}} \tag{4}$$

Note that in the above equation (4) only the numerator depends on the model. Henceforth, we will only consider the quantity in the numerator and we write it as

$$\bar{C}^{td}(m) = \sum_{j=1}^{N} \sum_{i=1}^{N} \mathbf{1}\{R^m(T_i, x_i) > R^m(T_i, x_j) \cdot \mathbf{1}\{T_j > T_i, D_i = m\}\} \tag{5}$$

The above equation can be simplified as

$$\bar{C}^{td}(m) = \sum_{i=1}^{|X^m|} \mathbf{1}\{R^m(T_{i,left}, X_i^m(left)) > R^m(T_{i,left}, X_i^m(right))\} \tag{6}$$

where $\mathbf{1}(x)$ is the indicator function, $X_i^m(left)$ $(X_i^m(right))$ is the left (right) element of the $i^{th}$ comparable pair in the set $X^m$ and $T_{i,left}$ $(T_{i,right})$ is the respective time-to-event. In the next section, we will use the above simplification (6) to construct the loss function for the neural network.

## 3    SIAMESE SURVIVAL PROGNOSIS NETWORK

In this section, we will describe the architecture of the network and the loss functions that we propose to train the network.

Denote $H$ as a feed-forward neural network which is visualized in figure 1. It is composed of a sequence of $L$ fully connected hidden layers with "scaled exponential linear units" (SELU) activation. The last hidden layer is fed to $M$ layers of width $K$. Each neuron in the latter $M$ layers estimates the probability that a subject $x$ experiences cause $m$ occurs in a time interval $t_k$, which is given as $Pr^m(t_k, x)$. For an input covariate $x$ the output from all the neurons is a vector of probabilities given as $\left\{ \left[ Pr^m(t_k, x) \right]_{k=1}^{K} \right\}_{m=1}^{M}$. The estimate of cumulative incidence function computed for cause $m$ at time $t_k$ is given as $\tilde{R}^m(t_k, x) = \sum_{i=1}^{k} Pr^m(t_i, x)$. The final output of the neural network for input $x$ is vector of estimates of the cumulative incidence function given as $H(x) = \left\{ \left[ \tilde{R}^m(t_k, x) \right]_{k=1}^{K} \right\}_{m=1}^{M}$.

In this section, we describe the loss functions that are used to train the network. The loss function is composed of three terms: discrimination, accuracy, and a loss term.

We cannot use the metric in (6) directly to train the network because it is a discontinuous function (composed of indicators) and this can lead to poor training of the network. We overcome this problem by approximating the indicator function using a scaled sigmoid function $\sigma(\alpha x) = \frac{1}{1+exp(-\alpha x)}$. The approximated discrimination index is given as

$$\hat{\tilde{C}}^{td}(m) \cong \tilde{C}^{td}(m) = \sum_{i=1}^{|X^m|} \sigma\Big[\alpha\big[\tilde{R}^m(T_{i,left}, X_i^m(left)) - \tilde{R}^m(T_{i,left}, X_i^m(right))\big]\Big] \tag{7}$$

The scaling parameter $\alpha$ determines the sensitivity of the loss function to discrimination. If the value of $\alpha$ is high, then the penalty for error in discrimination is also very high. Therefore, higher values of alpha guarantee that the subjects in a comparable pair are assigned concordant risk values.

The discrimination part defined above captures a model's ability to discriminate subjects for each cause separately. We also need to ensure that the model can predict the cause accurately. We define the accuracy of a model in terms of a scaled sigmoid function with scaling parameter $\kappa$ as follows:

$$Acc = \sum_{i=1}^{|X^m|} \sigma\Big[\kappa\big(\tilde{R}^{D_{left}}(T_{i,left}, X_i^m(left)) - \sum_{m \neq D_{left}} \tilde{R}^m(T_{i,left}, X_i^m(left))\big)\Big] \tag{8}$$

The accuracy term penalizes the risk functions only at the event times of the left subjects in comparable pairs. However, it is important that the neural network is optimized to produce risk values that interpolate well to other time intervals as well. In order to this, we introduce a loss term below.

$$Loss = \beta \sum_{m=1}^{M} \sum_{i=1}^{|X^m|} \sum_{t_k < T_{i,left}} R^m(t_k, X_i^m(right))^2 \tag{9}$$

The loss term ensures that the risk of each right subject is kept to as small a value as possible for all the times before time-to-event of the left subject in the respective comparable pair. Intuitively, the loss term can be justified as follows. The right subjects do not experience an event before the time $T_{i,left}$. Hence, the probability that they experience an event before $T_{i,left}$ should take a small value.

The final loss function is the sum of the discrimination terms (described above), the accuracy and the loss terms, and is given as

$$\sum_{m=1}^{M} \hat{\bar{C}}^{td}(m) + Acc + Loss \qquad (10)$$

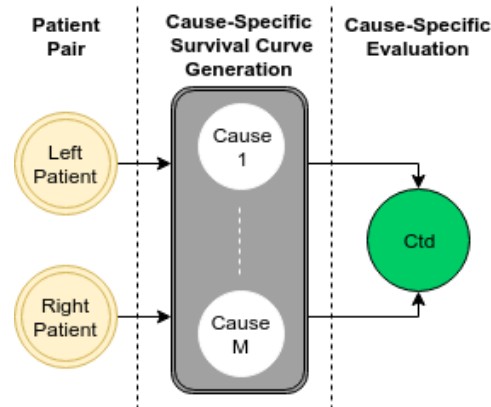

Figure 1: Illustration of the architecture.

Finally, we adjust for the event imbalance and the time interval imbalance caused by the unequal number of pairs for each event and time interval with inverse propensity weights. These weights are the frequency of the occurrence of the various events at the various times and are multiplying the loss functions of the corresponding comparable pairs.

We train the feed-forward network using the above loss function (10) and regularize it using SELU dropout (Klambauer et al. (2017)). Since the loss function involves the discrimination term, each term in the loss function involves a pairwise comparison. This makes the network training similar to a Siamese network (Bromley et al. (1994)). The backpropagation terms now depend on each comparable pair.

## 4 EXPERIMENTS

This section includes a discussion of hyper-parameter optimization which is followed by competing risk and survival analysis experiments. For single event problems we compare against Cox PH model ("survival" R package), (Katzman et al. (2016)) (github) and Survival Random Forest ("randomForestSRC" R package). For competing risk problems we compare against Fine-Gray model ("cmprsk" R package), Competing Random Forest ("randomForestSRC" R package) and the cause-specific (cs) extension of two single event (non-competing risks) methods, Cox PH model and (Katzman et al. (2016)). In cause-specific extension of single event models, we mark the occurrence of any event apart from the event of interest as censorship and decouple the problem into separate single event problem (one for each cause); this is a standard way of extending single-event models to competing risk models. In the following results we refer to our method with the acronym SSPN.

Table 1: Summary of hyper-parameters

| Parameter | batch size | # hidden layers | hidden layers width | dropout rate |
|---|---|---|---|---|
| SEER | 2048 | 3 | 50 | 0.4 |
| Synthetic data | 2048 | 2 | 40 | 0.4 |
| UNOS | 4096 | 3 | 40 | 0.4 |
| UK Biobank | 1024 | 3 | 30 | 0.3 |

### 4.1 HYPER-PARAMETER OPTIMIZATION

Optimization was performed using a 5-fold cross validation with fixed censorship rates in each fold. We choose 60-20-20 division for training, validation and testing sets. A standard grid search was used to determine the batch size, the number of hidden layers and the width of the hidden layers and the dropout rate. The optimal values of $\alpha$ and $\beta$ were consistently 500 and 0.01 for all datasets.

As previously mentioned, the sets are comprised of patient pairs. In each training iteration, a batch size of pairs was sampled with replacement from the training set which reduces convergence speed but doesn't lower performance relative to the standard non replacement sampled batches (Recht & Re (2012)). The validation performance was measured every 1000 training iterations during which the stopping criterion conditions were evaluated. We defined the stopping criterion as the lack of validation improvement in terms of our metric on all of the causes over the last x evaluations. We note that the training sets are commonly in the tens of million pairs with patients appearing multiple times in both sides of the pair. A standard definition of an epoch would compose of a single iteration over all patient. However, in our case, we not only learn patient specific characteristics but also patient comparison relationships, which means an epoch with number of iterations equal to the number of patients is not sufficient. On the other hand, an epoch definition as an iteration over all pairs is impractical. Our best empirical results were attained after 100K iterations with Tensorflow on 8-core Xeon E3-1240 with 32GB Ram The usage of SELU activation and dropout followed confirmation of its superiority over ReLU on the tested datasets. We note that ReLU activation generated similar performance gains over the benchmarks although lesser than SELU. We used SELU weight initialization, $\mathcal{N}(0, input_size^{-1})$, Adam optimizer (Kingma & Ba (2014)) and a decaying learning rate, $LR^{-1}(i) = LR(0) + i, LR(0) = 0.001$. Table 1 summarizes the different optimized hyper-parameters.

Table 2: Summary of competing $C^{td}$ index on SEER.

| Dataset | CVD | Breast Cancer | Other |
|---|---|---|---|
| cs-Cox PH | 0.656 [0.629-0.682] | 0.634 [0.626-0.642] | 0.695 [0.675-0.714] |
| cs-(Katzman et al. (2016)) | 0.645 [0.625-0.664] | 0.697 [0.686-0.708] | 0.675 [0.644-0.706] |
| Fine-Gray | 0.659 [0.605-0.714] | 0.636 [0.622-0.650] | 0.691 [0.673-0.708] |
| Competing Random Forest | 0.601 [0.565-0.637] | 0.705 [0.692-0.718] | 0.636 [0.624-0.648] |
| SSPN | **0.663 [0.625-0.701]** | **0.735 [0.678-0.793]** | **0.699 [0.681-0.716]** |

*p-value $< 0.05$

## 4.2 SEER

The Surveillance, Epidemiology, and End Results Program (SEER)[1] dataset provides information on breast cancer patients during the years 1992-2007. A total of 72,809 patients, experienced breast cancer, cardiovascular disease (CVD), other diseases, or were right-censored. The cohort consists of 23 features, including age, race, gender, morphology information, diagnostic information, therapy information, tumor size, tumor type, etc. Missing values were replaced by mean value for real-valued features and by the mode for categorical features. 1.3% of the patients experienced CVD and 15.6% experienced breast cancer. Table 2 displays the results for this dataset. We can notice that for the infrequent adverse event, CVD, the performance gain is negligible while for the frequent breast cancer event, the gain is significant.

## 4.3 SYNTHETIC DATA

Due to the relative scarcity of competing risks datasets and methods, we have created an additional synthetic dataset to further demonstrate the performance of our method. We have constructed two stochastic processes with parameters and the event times as follows:

$$x_i^1, x_i^2, x_i^3 \sim \mathcal{N}(0, \mathbf{I}), T_i^1 \sim \exp\left((x_i^3)^2 + x_i^1\right), T_i^2 \sim \exp\left((x_i^3)^2 + x_i^2\right) \quad (11)$$

where $(x_i^1, x_i^2, x_i^3)$ is the vector of features for patient $i$. For $k = 1, 2$, the features $x^k$ only have an effect on the event time for event $k$, while $x^3$ has an effect on the event times of both events. Note that we assume event times are exponentially distributed with a mean parameter depending on both linear and non-linear (quadratic) function of features. Given the parameters, we first produced $30,000$ patients; among those, we randomly selected $15,000$ patients (50%) to be right-censored at a time randomly drawn from the uniform distribution on the interval $[0, \min\{T_i^1, T_i^2\}]$. (This

---

[1] https://seer.cancer.gov/causespecific/

censoring fraction was chosen to be roughly the same censoring fraction as in the real datasets, and hence to present the same difficulty as found in those datasets.) Table 3 displays the results for the above dataset. We can notice the same consistent performance gain as in the previous case.

Table 3: Summary of competing $C^{td}$ index on synthetic data.

| Method | Cause 1 | Cause 2 |
|---|---|---|
| cs-Cox PH | 0.571 [0.554-0.588] | 0.581 [0.570-0.591] |
| cs-(Katzman et al. (2016)) | 0.580 [0.556-0.603] | 0.593 [0.576-0.611] |
| Fine-Gray | 0.574 [0.559-0.590] | 0.586 [0.577-0.594] |
| Competing Random Forest | 0.591 [0.575-0.606] | 0.573 [0.557-0.588] |
| SSPN | **0.603 [0.593-0.613]** | **0.613 [0.598-0.627]** |

*p-value $< 0.05$

## 5 CONCLUSION

Competing risks settings are ubiquitous in medicine. They can be encountered in cardiovascular diseases, in cancer, and in the geriatric population suffering from multiple diseases. To solve the challenging problem of learning the model parameters from time-to-event data while handling right censoring, we have developed a novel deep learning architecture for estimating personalized risk scores in the presence of competing risks which is based on the well-known Siamese network architecture. Our method is able to capture complex non-linear representations missed out by classical machine learning and statistical models. Experimental results show that our method is able to outperform existing competing risk methods by successfully learning representations which can flexibly describe non-proportional hazard rates with complex interactions between covariates and survival times that are common in many diseases with heterogeneous phenotypes.

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

## A    SURVIVAL ANALYSIS RESULTS

### A.1    UNOS

The United Network for Organ Sharing (UNOS) database[2] consists of patients who underwent heart transplantation in the period 1985-2015. Of the total of 60,400 patients who received heart transplants, 29,436 patients (48.7%) were followed until death; the remaining 30,964 patients (51.3%) were right-censored. A total of 50 features (30 recipient-relevant, 9 donor-relevant and 11 donor-recipient compatibility) were used. Table 4 presents the results for the UNOS dataset. The results show clear performance improvements for the Siamese Survival Prognosis Network.

Table 4: Summary of survival $C^{td}$ index.

| Method | UNOS | UK Biobank |
|---|---|---|
| Cox PH | 0.564 [0.558-0.570] | 0.743 [0.731-0.754] |
| (Katzman et al. (2016)) | 0.576 [0.550-0.601] | 0.693 [0.651-0.734] |
| Survival Random Forest | 0.577 [0.571-0.582] | 0.686 [0.674-0.699] |
| SSPN | **0.594 [0.576-0.611]** | **0.748 [0.723-0.774]** |

*p-value $< 0.05$

### A.2    UK BIOBANK

UK Biobank is a comprehensive dataset consisting of health records, diagnoses and treatments of a wide array of diseases, including Cardiovascular disease (CVD) events. There is a total of 413,119 patients, followed for 10 years, with no previous history of CVD, out of whom 6,051 (1.5%) developed a CVD. The records consist of 8 covariates (gender, age, smoking habits, systolic blood pressure, blood pressure treatment, total cholesterol, HDL cholesterol and diabetes). Similarly to UNOS, the results in table 4 show clear performance improvements for the Siamese Survival Prognosis Network.

---

[2]https://www.unos.org/data/

