# OpenReview forum: "Siamese Survival Analysis with Competing Risks"
_ICLR.cc/2018/Conference — Reject_

### Official Review · AnonReviewer1 · 2017-11-27
**Re: Siamese survival analysis**

**Rating:** 4
**Confidence:** 4

**Review:**

This paper introduces siamese neural networks to the competing risks framework of Fine and Gray. The authors optimize for the c-index by minimizing a loss function driven by the cumulative risk of competing risk m and correct ordering of comparable pairs. While the idea of optimizing directly for the c-index directly is a good one (with an approximation and with useful complementary loss function terms), the paper leaves something to be desired in quality and clarity.

Related works:
- For your consideration: is multi-task survival analysis effectively a competing risks model, except that these models also estimate risk after the first competing event (i.e. in a competing risks model the rates for other events simply go to 0 or near-zero)? Please discuss. Also, if the claim is that there are not deep learning survival analyses, please see, e.g. Jing and Smola.
- It would be helpful to define t_k explicitly to alleviate determining whether it is the interval time between ordered events or the absolute time since t_0 (it's the latter). Consider calling k a time index instead of t_k a time interval ("subject x experiences cause m occurs [sic] in a time interval t_k")
- Line after eq 8: do you mean accuracy term?
- I would not call Reg a regularization term since it is not shrinking the coefficients. It is a term to minimize a risk not a parameter.
- You claim to adjust for event imbalance and time interval imbalance but this is not mathematically shown nor documented in the experiments.
- The results show only one form of comparison, and the results have confidence intervals that overlap with at least one competing method in all tasks.

---

> ### Author Response · Authors · 2018-01-04
> **Please see the revised paper**
>
> Q1. the paper leaves something to be desired in quality and clarity.
>
> A1. We worked hard to improve the paper’s clarity. We thank all the reviewers for their valuable comments which helped us in this pursuit and we are hopeful that the reviewers will reconsider their scores after seeing the revised paper.
>
>
> Q2. For your consideration: is multi-task survival analysis effectively a competing risks model, except that these models also estimate risk after the first competing event (i.e. in a competing risks model the rates for other events simply go to 0 or near-zero)? Please discuss.
>
> A2. Multi-task survival analysis can be indeed interpreted as a competing risks model. However, most works on competing risks assume that each subject experienced only a single event (See for instance the state-of-the-art Fine Gray model). This assumption originates from the constraints posed by actual clinical data (such as the well-known SEER dataset) where risks commonly correspond to deaths from various causes.
>
> .
> Q3. Also, if the claim is that there are not deep learning survival analyses, please see, e.g. Jing and Smola.
>
> A3. We do not aim to make such claims regarding the classical single risk survival analysis.
> In fact, we compare against such conventional survival analysis benchmarks such as the deep learning survival analysis algorithm in [18]. Our paper only claimed that this is the first deep learning architecture for survival analysis in the presence of competing risks (please refer to A1 reviewer 1 for the differences between the problems). We are sorry for the confusion caused and have now made clear our contributions in the revised paper.
>
>
> Q4. It would be helpful to define t_k explicitly to alleviate determining whether it is the interval time between ordered events or the absolute time since t_0 (it's the latter). Consider calling k a time index instead of t_k a time interval ("subject x experiences cause m occurs [sic] in a time interval t_k")
>
> A4. We have made the change. Thank you.
>
>
> Q5. Line after eq 8: do you mean accuracy term?
>
> A5. We renamed this term and thank the reviewer again.
>
>
> Q6. I would not call Reg a regularization term since it is not shrinking the coefficients. It is a term to minimize a risk not a parameter.
>
> A6. We will rename this term as a loss term.
>
>
> Q7. You claim to adjust for event imbalance and time interval imbalance but this is not mathematically shown nor documented in the experiments.
>
> A7. We adjust for event imbalance and time interval imbalance using inverse propensity weights. These weights are the frequency of the occurrence of the various events at the various times. We have now clarified this point in the revised paper.
>
>
> Q8. The results show only one form of comparison, and the results have confidence intervals that overlap with at least one competing method in all tasks.
>
> A8. We have optimized the time-dependent discrimination index. If we were to optimize a different evaluation metric, we would include a different form of comparison. We agree that some confidence intervals overlap, however, this fact does not contradict the claim that this paper succeeds in providing a statistically significant improvement over the state-of-the-art on all datasets. Non-overlapping confidence intervals are a sufficient but not a necessary condition for  statistical significance.
>
> References
>
> [18] Katzman, Jared, et al. "Deep survival: A deep cox proportional hazards network." arXiv preprint arXiv:1606.00931 (2016).

---

### Official Review · AnonReviewer3 · 2017-11-28
**Modest technical contribution, minor gains in performance with competitors**

**Rating:** 4
**Confidence:** 4

**Review:**

The authors tackle the problem of estimating risk in a survival analysis setting with competing risks. They propose directly optimizing the time-dependent discrimination index using a siamese survival network. Experiments on several real-world dataset reveal modest gains in comparison with the state of the art.

- The authors should clearly highlight what is their main technical contribution. For example, Eqs. 1-6 appear to be background material since the time-dependent discrimination index is taken from the literature, as the authors point out earlier. However, this is unclear from the writing.

- One of the main motivations of the authors is to propose a model that is specially design to avoid the nonidentifiability issue in an scenario with competing risks. It is unclear why the authors solution is able to solve such an issue, specially given the modest reported gains in comparison with several competitive baselines. In other words, the authors oversell their own work, specially in comparison with the state of the art.

- The authors use off-the-shelf siamese networks for their settting and thus it is questionable there is any novelty there. The application/setting may be novel, but not the architecture of choice.

- From Eq. 4 to Eq. 5, the authors argue that the denominator does not depend on the model parameters and can be ignored. However, afterwards the objective does combine time-dependent discrimination indices of several competing risks, with different denominator values. This could be problematic if the risks are unbalanced.

- The competitive gain of the authors method in comparison with other competing methods is minor.

- The authors introduce F(t, D | x) as cumulative incidence function (CDF) at the beginning of section 2, however, afterwards they use R^m(t, x), which they define as risk of the subject experiencing event m before t. Is the latter a proxy for the former? How are they related?

---

> ### Author Response · Authors · 2018-01-04
> **Please see the revised paper**
>
> Q1. The authors should clearly highlight what is their main technical contribution. For example, Eqs. 1-6 appear to be background material since the time-dependent discrimination index is taken from the literature, as the authors point out earlier. However, this is unclear from the writing.
>
> A1. We agree with the reviewer that the main technical contributions were not clearly stated. We have now improved the paper and clearly highlight our main technical contributions as follows:
>
> - We develop a novel Siamese feed-forward neural network for survival analysis with competing risks. Our novel neural network architecture is designed to optimize concordance. This is achieved by estimating risks in a relative fashion, meaning, the risk for the “true” event of a patient (i.e. the event which actually took place) must be higher than:
> 1. all other risks for the same patient (Eq. 8);
> 2. the risks for the same true event of other patients that experienced it at a later time (Eq. 7).
>
> - Because our neural network issues a joint risk for all competing events, our architecture needs to compare different risks for the different events at different times and arrange them in a concordant fashion (earlier time means higher risk for any pair of patients). To enable such a comparison, we develop a novel type of Siamese network architecture. Unlike previous Siamese neural networks architectures [14]-[16] which were developed for different purposes such as learning the pairwise similarity between different inputs, our architecture aims to maximize the gap between output risks among the different inputs. Instead of learning a representation that captures the similarities between the inputs, we learn a representation that generates the highest possible difference between the outputs.
> By estimating the risks of all causes jointly, our Siamese survival network for competing risks is able to learn a shared representation that captures the latent structure of the data and allows estimating cause-specific risks.
>
> - We use a loss term (Eq. 9) that is based on the structure of the comparable pairs (the right patient has a longer event time). This component comes in the form of a learning constraint; that is, the survival curve of the right patient (longer survival time) must be lower than the survival curve of the left patient (shorter survival time) up to the event time of the left patient. This improves the generalization capabilities of our algorithm.
>
> We have now clarified in the revised paper that equations 1-3 define the well-known time-dependent discrimination index and equation 4 is an estimator for it, as given in [17].
> Equations 5 and 6 are simplifications of the above index that we present before introducing our approximation. We have added the above citation before equations 1 and 4 for the sake of clarity.
>
>
> Q2. One of the main motivations of the authors is to propose a model that is specially design to avoid the nonidentifiability issue in an scenario with competing risks. It is unclear why the authors solution is able to solve such an issue,
>
> A2. Nonidentifiability in the competing risks settings arises from the inability to estimate the true cause-specific survival curves from the empirical data. However, this paper focuses on generating concordant risks in a relative fashion as opposed to true cause-specific survival curves. By avoiding the  estimation of the true cause-specific survival curves, we are able to avoid the nonidentifiability problem.
>
>
> Q3. specially given the modest reported gains in comparison with several competitive baselines. In other words, the authors oversell their own work, specially in comparison with the state of the art.
>
> A3. Our paper provides a statistically significant improvement over the state-of-the-art methods on survival analysis with competing risks on both synthetic as well as on real medical data. We wish to stress that our focus is on the medical domain were even minor gains are important because of the potential to save lives. For example, there are 72809 patients in the SEER dataset we used. A performance improvement even as low as 0.1% has the potential to save lives and therefore should not be disregarded. However, we tamed our claims in the revised paper and explained better why we believe the gains obtained matter.
>
>
> Q4. The authors use off-the-shelf siamese networks for their setting and thus it is questionable there is any novelty there. The application/setting may be novel, but not the architecture of choice.
>
> A4. We are very sorry for not clearly describing the novelty of the proposed architecture. We have now clarified the architectural novelty. Please refer to A1.

---

> > ### Author Response · Authors · 2018-01-04
> > **Continue**
> >
> > Q5. From Eq. 4 to Eq. 5, the authors argue that the denominator does not depend on the model parameters and can be ignored. However, afterwards the objective does combine time-dependent discrimination indices of several competing risks, with different denominator values. This could be problematic if the risks are unbalanced.
> >
> > A5. We agree with the reviewer that in the case of unbalanced risks, the denominators of different discrimination indices cannot be ignored. We overcome this by balancing the risks using inverse propensity weighting. Please refer to the sentence following equation 10:
> > “Finally, we adjust for the event imbalance and the time interval imbalance caused by the unequal number of pairs for each event and time interval with inverse propensity weights on the loss function.”
> >
> >
> > Q6.  The competitive gain of the authors method in comparison with other competing methods is minor.
> >
> > A6. As mentioned before, since our method is on medical applications, where these methods can be used for improving outcomes or providing interventions, even relatively small performance improvements can lead to improved healthcare delivery. Our method is especially suitable for dealing with competing risks in multi-morbid population which represents a major healthcare challenge. Multimorbidity – the accumulation of multiple chronic diseases – has emerged as a major contemporary challenge of the ageing population. More than two-thirds of people aged over 65 are nowadays multimorbid, i.e. have two or more chronic diseases. However, current prognosis models are not designed to consider diseases in combination leading to poor use of scant resources and complications. Our method is one of the few methods especially designed to address this important problem. Thus, a performance improvement even as low as 0.1% has the potential to save many lives (since the majority of the elderly population is multimorbid) and improve healthcare delivery and utilization.
> >
> >
> > Q7. The authors introduce F(t, D | x) as cumulative incidence function (CDF) at the beginning of section 2, however, afterwards they use R^m(t, x), which they define as risk of the subject experiencing event m before t. Is the latter a proxy for the former? How are they related?
> >
> > A7.  As the reviewer pointed out, Rm(t,x) is indeed a proxy for the CDF of cause m, F(t,D|x).
> > We deem the R notation necessary since it is used to symbolize the algorithm’s output. We have that F(t,D=m|x)=Rm(t,x).
> >
> > References
> >
> > [14] Bromley, Jane, et al. "Signature verification using a" siamese" time delay neural network." Advances in Neural Information Processing Systems. 1994.
> >
> > [15] Chopra, Sumit, Raia Hadsell, and Yann LeCun. "Learning a similarity metric discriminatively, with application to face verification." Computer Vision and Pattern Recognition, 2005. CVPR 2005. IEEE Computer Society Conference on. Vol. 1. IEEE, 2005.
> >
> > [16] Wang, Juan, et al. "A multi-resolution approach for spinal metastasis detection using deep Siamese neural networks." Computers in Biology and Medicine 84 (2017): 137-146.
> >
> > [17] Antolini, Laura, Patrizia Boracchi, and Elia Biganzoli. "A time‐dependent discrimination index for survival data." Statistics in medicine 24.24 (2005): 3927-3944.

---

### Official Review · AnonReviewer2 · 2017-11-29
**Brittle application of deep learning to competitive risk analysis.**

**Rating:** 4
**Confidence:** 5

**Review:**

The paper entitled 'Siamese Survival Analysis' reports an application of a deep learning to three cases of competing risk survival analysis. The author follow the reasoning that '... these ideas were not explored in the context of survival analysis', thereby disregarding the significant published literature based on the Concordance Index (CI).

Besides this deficit, the paper does not present a proper statistical setup (e.g. 'Is censoring assumed to be at random? ...) , and numerical results are only referring to some standard implementations, thereby again neglecting the state-of-the-art solution. That being said, this particular use of deep learning in this context might be novel.

---

> ### Author Response · Authors · 2018-01-04
> **Please see the revised paper**
>
> Q1. The paper entitled 'Siamese Survival Analysis' reports an application of a deep learning to three cases of competing risk survival analysis. The author follow the reasoning that '... these ideas were not explored in the context of survival analysis', thereby disregarding the significant published literature based on the Concordance Index (CI).
>
> A1. We are sorry for the confusion which the misnaming of our method has caused. We should have dubbed our method “survival analysis with competing risks” rather than simply “survival analysis”. The area of survival analysis with competing risks is a much less explored research domain. Survival analysis with competing risks differs from the single risk survival analysis for several reasons:
>
> - The times to the different event types (causes) are generally not independent [2]-[4]. As a result, our analysis requires developing a joint estimation model that can account for the latent relationships. (See for instance one of the earliest works on this problem [1] and more recent studies [2]-[4].)
>
> - We acknowledge that a significant amount of work was dedicated to c-index optimization for survival analysis. However, the problem of competing risks is much less studied. This is partly due to the fact that the c-index is designed for the single risk setting; competing risks problems require a different optimization metric. Therefore, our algorithm optimizes a time-dependent discrimination index for competing risks. This index cannot be optimized (in a straightforward manner) using the algorithms in the state-of-the-art survival analysis works. For instance, a smooth approximation of the c-index was optimized in [5]-[7] by using gradient boosting. These works require a smooth approximation of the time-dependent discrimination index and derivatives of it (for gradient boosting), which are challenging to obtain; thus, extensions of these methods to our considered problem are not straightforward. In [8], a random forest for survival analysis with competing risks was grown using the c-index as the splitting rule. Also in this case, the extension to the time-dependent discrimination index for competing risks as the splitting rule is not straightforward.
>
>
> Q2. Besides this deficit, the paper does not present a proper statistical setup (e.g. 'Is censoring assumed to be at random? ...)
>
> A2. This paper makes standard assumptions that are commonly used in survival analysis [1]-[8]. Specifically, censoring is assumed to be independent, i.e., the time-to-event conditional on covariates is independent of other variables including censoring [9]-[12]. We have clarified these assumptions in the revised paper. Thank you.
>
>
> Q3. and numerical results are only referring to some standard implementations, thereby again neglecting the state-of-the-art solution.
>
> A3. Again, we are very sorry for the confusion created. Our paper primarily focuses on the problem of survival analysis with competing risks, where only a limited number of solutions exist: the Fine-Gray model [9] and the Competing Risks Forest [13]. We have compared their method with both algorithms on multiple datasets and showed a consistent improvement as seen in tables 2 and 3. Table 4 illustrated results for the conventional single risk survival analysis and serves only as a sanity check for the competing risks benchmarks. To remove future confusion about the focus of this paper, we have moved this table to the appendix.
>
>
> Q4. That being said, this particular use of deep learning in this context might be novel.
>
> A4. We thank you for this acknowledgment and we hope the reviewer will reconsider his/her score following the revised version of this paper.

---

> > ### Author Response · Authors · 2018-01-04
> > **References**
> >
> > [1] Elandt-Johnson, Regina C. "Conditional failure time distributions under competing risk theory with dependent failure times and proportional hazard rates." Scandinavian Actuarial Journal 1976.1 (1976): 37-51.
> >
> > [2] Lim, Hyun J., et al. "Methods of competing risks analysis of end-stage renal disease and mortality among people with diabetes." BMC medical research methodology 10.1 (2010): 97.
> >
> > [3] Lambert, P. C., et al. "Estimating the crude probability of death due to cancer and other causes using relative survival models." Statistics in medicine 29.7‐8 (2010): 885-895.
> >
> > [4] Satagopan, J. M., et al. "A note on competing risks in survival data analysis." British journal of cancer 91.7 (2004): 1229-1235.
> >
> > [5] Chen, Yifei, et al. "A gradient boosting algorithm for survival analysis via direct optimization of concordance index." Computational and mathematical methods in medicine 2013 (2013).
> >
> > [6] Mayr, Andreas, and Matthias Schmid. "Boosting the concordance index for survival data–a unified framework to derive and evaluate biomarker combinations." PloS one 9.1 (2014): e84483.
> >
> > [7] Mayr, Andreas, Benjamin Hofner, and Matthias Schmid. "Boosting the discriminatory power of sparse survival models via optimization of the concordance index and stability selection." BMC bioinformatics 17.1 (2016): 288.
> >
> > [8] Schmid, Matthias, Marvin N. Wright, and Andreas Ziegler. "On the use of Harrell’s C for clinical risk prediction via random survival forests." Expert Systems with Applications 63 (2016): 450-459.
> >
> > [9] Fine, Jason P., and Robert J. Gray. "A proportional hazards model for the subdistribution of a competing risk." Journal of the American statistical association 94.446 (1999): 496-509.
> >
> > [10] Crowder, Martin J. Classical competing risks. CRC Press, 2001.
> >
> > [11] Gooley, Ted A., et al. "Estimation of failure probabilities in the presence of competing risks: new representations of old estimators." Statistics in medicine 18.6 (1999): 695-706.
> >
> > [12] Tsiatis, Anastasios. "A nonidentifiability aspect of the problem of competing risks." Proceedings of the National Academy of Sciences 72.1 (1975): 20-22.
> >
> > [13] Ishwaran, Hemant, et al. "Random survival forests for competing risks." Biostatistics 15.4 (2014): 757-773.

---

### Decision · Program_Chairs · 2018-01-29
**ICLR 2018 Conference Acceptance Decision**

**Decision:**

Reject

**Comment:**

Reviewers unanimous in assessment that manuscript has merits, but does not satisfy criteria for publication.

Pros:
- Potentially novel application of neural networks to survival analysis with competing risks, where only one terminal event from one risk category may be observed.

Cons:
- Incomplete coverage of other literature.
- Architecture novelty may not be significant.
- Small performance gains (though statistically significant)